# Detection Technique of Partial Discharges at Impulse Voltage with Appropriate Filter Settings for Signal Separation

**Karsten Backhaus** [1,*,†] **, Lena Elspaß** [1,†] **and Konstantin Pasche** [2]

[1] Institute of Electrical Power Systems and High Voltage Technology, Technische Universität Dresden, 01062 Dresden, Germany; lena.elspass@tu-dresden.de

[2] Intitute of Electrical Power Engineering, Technische Universtiät Dresden, 01062 Dresden, Germany; konstantin.pasche@tu-dresden.de

*\** Correspondence: karsten.backhaus@tu-dresden.de

† These authors contributed equally to this work.

**Abstract:** Due to the increased utilization of electric converters feeding rotating high voltage motors, their insulation is subject to transient impulse and high frequency oscillating voltages. In corresponding life time experiments with repetitive oscillating impulse voltage at winding insulation samples, higher life time coefficients were observed than known from previous investigations and operational experience. In order to understand the discharge and aging phenomena, the purpose of this work is the secure detection of partial discharges in solid and solid–air insulation types for transient impulse voltage stress by applying an adequate partial discharge (PD) measurement technique to future life time experiments. It is shown that partial discharges under impulsive voltages can be detected with conventional measuring equipment using broadband shunts, as well as inductive antennas. It becomes apparent that a precise voltage source, a precise shunt, as well as a high resolution oscilloscope are mandatory for reliable current measurement results. As a part of the analysis of the measurement data, it is shown that partial discharges can be distinguished from the displacement current caused by impulse voltages in a capacitive insulation material, as well as noise and disturbance from the measurement environment. As a first approach, a high order bandpass filter is applied in order to gain sound signals for future automated signal separation.

**Keywords:** measurement techniques; detection; partial discharges; converter fed motors; PWM; impulse voltage; disturbance; noise; filter; separation

## 1. Introduction

Due to the increased usage of power electronic components, rotating high voltage machines are more often subject to transient voltage stresses with a high repetition frequency. These can cause partial discharges (PD) in the different insulation of motors: phase, main, and winding insulation. For steady PD in terms of repetition rate or respectively total number directly linked to the end-of-life of the insulation systems [1], it is essential to be able to determine their occurrence in converter fed machines and in life time experiments to evaluate new insulation systems.

Especially, PD that occur inside the insulation rather than on the surface are relevant as they advance the successive aging of the solid insulation significantly. The total number of PD, as well as the charge amount can be an indicator of the state of the insulation as it is related to the growth of a discharge channel [2]. Thus, it is important to detect PD in their quantity correctly. The author in [1] concluded that the life time curves of one insulation system determined with AC voltage and impulse voltage will differ by time in regard to the life time exponent, but they can be compared by taking the

number sine halfwaves and impulse. The physical basis for this conclusion is the repetitive PD events for every sine half wave or impulse, respectively.

The author of [3] on the contrary determined life time curves with even a steeper life time coefficient for a high frequency damped oscillating impulse. The cause was seen as a potentially higher occurrence of PD, which cannot be measured so far by the authors. Damped oscillating impulses are seen as an ideal test voltage form to investigate the electric behavior of insulation systems for converter driven machines, as they are stressed both by high dielectric losses and PD occurrence. In contrast to the square impulse, the bipolar damped impulses deliver quicker test results due to the higher number of zero crossings or trains of $U_{pp}$-events in accordance with IEC TS 60034-27-5 [4]. It is also the more realistic wave shape compared to the stress in a converter driven motor.

In order to gain further knowledge about the aging mechanisms of solid insulation systems for impulse voltage stress in the future, this paper focuses on the electric measurement of PD under lightning impulse (LI) voltage stress according to IEC 60060-1 as an initial step to gain reproducible results. It is well known that a precise voltage form is essential for current measurements. The experience herewith collected will in the future be transferred to damped oscillating impulse life testing of solid machine insulation systems.

To enhance the confidence of the measurement results, simultaneous measurements on different physical bases are carried out:

(1) The electric current through the investigated specimen measured with a shunt;
(2) The voltage signal determined by a monopole antenna;
(3) The voltage signal determined by an orthogonal inductive sensor.

The insulation of rotating machines is of a capacitive nature. Thus, transient impulse voltage stress leads to a displacement current in the form of an impulse within the dielectric with a higher frequency content of about one order. If the voltage stress is higher than the inception voltage, PD can occur. Their current signal has an impulse shape as well. Both, the PD and the displacement current overlap in the shunt based current measurement. While the displacement current does not reflect the state of the insulation, PD in their quantity are an important measure. Thus, it is important to distinguish between both current components for the correct evaluation. The signal can be transferred into the frequency domain. There, the two components can be distinguished based on their amplitude spectrum, which correlates with the rise time of the impulses. The shorter the rise time, the higher are the frequency components.

## 2. PD-Measurement Methods

PD are accompanied by physical and electrical phenomena. Thus, electrical and non-electrical measurement methods exist [5]. Phenomena such as acoustic waves, light emission, or chemical decomposition products are used as non-electrical measurements. These allow the detection of PD under certain circumstances, but no quantification for the specimen like the solid insulation system of a test coil or a motorette. Electric measurements offer better possibilities for the quantification. The number of PD, as well as the apparent charge amount can be determined. Both allow conclusions on the state of the insulation. Generally, electric PD measurement can be divided into several categories, namely PD measurement in accordance with IEC 60270, HF/VHF (high frequencies/very high frequencies), and UHF (ultra high frequencies) measurements. The IEC 60270 covers PD measurements under AC stress of up to 400 Hz as well as measurements under DC stress. Figure 1 depicts the frequency ranges that are covered by the different categories. A typical amplitude spectrum of a PD impulse in a solid material [6] and of the corresponding displacement current as a result of an LI are displayed. Furthermore, two exemplary filters (Filter 1 with $f_1/f_2 = 100\,\text{kHz}/1\,\text{MHz}$ and Filter 2 with $f_1/f_2 = 585\,\text{kHz}/615\,\text{kHz}$) in accordance with IEC 60270 are included, which are used to display the influence of filter parameters in the following.

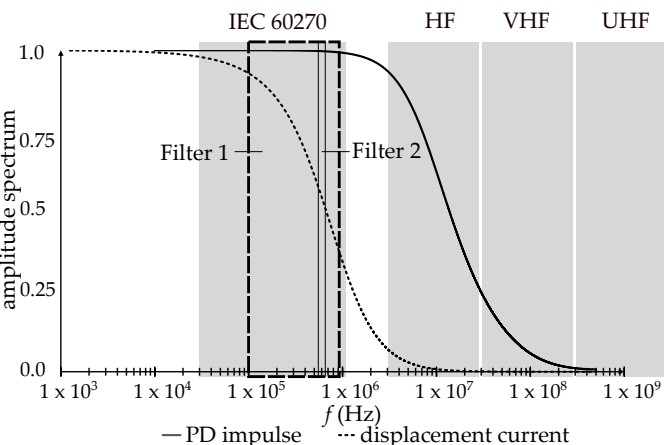

**Figure 1.** Amplitude spectrum of the displacement current in a solid insulation and of a PD impulse, as well as the ranges for PD measurements according to [7].

Figure 1 indicates why no conventional UHV system is chosen, as the signal content in this frequency range is very low. An amplifier with tremendous amplification ratio is required in order to achieve millivolt signals.

To evaluate the charge caused by PD, IEC 60270 suggests the so-called "quasi-integration". That means applying a bandpass filter to the measured current within the range of the constant amplitude spectrum. Then, the amplitude of the response signal is directly proportional to the charge. Within the frequency range that is suggested in IEC 60270, the amplitude spectrum of the PD, as well as the displacement current overlap significantly. Thus, the frequency range is not eligible for PD measurements based on quasi-integration under transient voltage stress, but higher frequencies should be considered.

IEC 60270 differentiates between two kinds of bandpass filters: broadband and narrow band. Broadband refers to a bandwidth that is higher than the lower cut-off frequency ($f_1 = (30..100)$ kHz, $f_2 \leq 1$ MHz, $\Delta f = (100...900)$ kHz). The bandwidth of a narrow band filter is significantly smaller ($f_\mathrm{m} = (50...1000)$ kHz, $\Delta f = (9..30)$ kHz). Applying a filter to a signal has a significant influence on the signal (Figure 2), as the result of a MATLAB calculation shows. For one, the amplitude of the signals is reduced significantly, here by a factor of more than 100. Furthermore, the bandwidth of a filter has a significant influence on the filtered signal. A broadband filter results in a strongly damped oscillation (Figure 2b), while a narrow band filter results in a only slightly damped oscillation (Figure 2c).

It is apparent that both filters alter the input signal. When using a broadband filter, the main impulse stays clearly visible. When using a narrow band filter, the main impulse is no longer detectable, but rather, a stretched oscillation is apparent. Thus, a single impulse can be mistaken for several. Additionally, the duration of the filtered impulse increases significantly. That affects the ability to detect several separate impulses as the filtered signals are more likely to overlap the longer their duration is. Thus, PD measurements should be conducted using a broadband filter.

In the presented results, a high pass broadband filter of fifth order was applied in order to achieve a separation of current components caused by the impulse voltage and the PD.

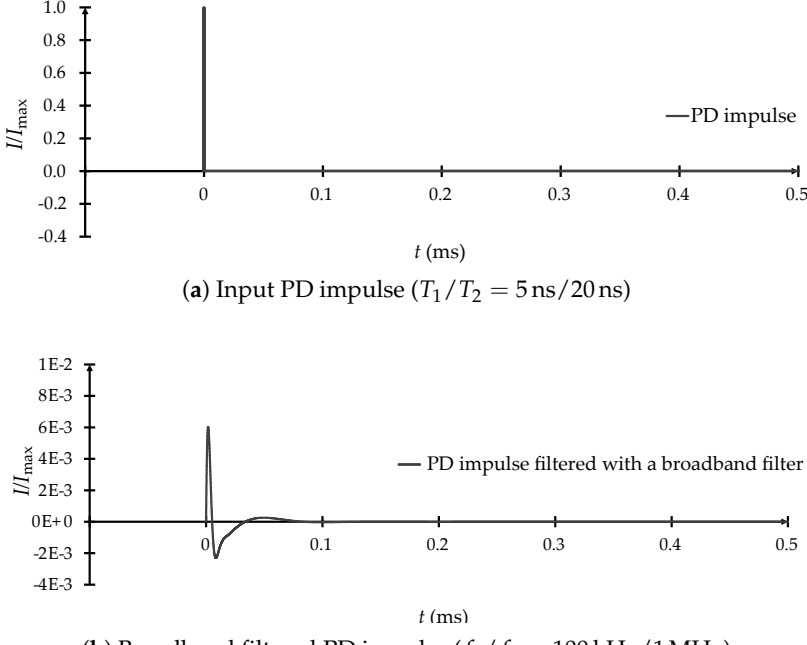

(**a**) Input PD impulse ($T_1/T_2 = 5\,\text{ns}/20\,\text{ns}$)

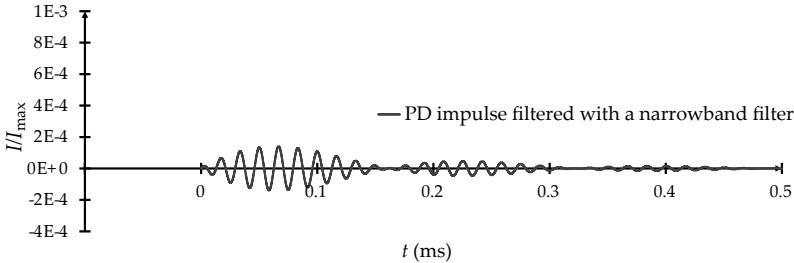

(**b**) Broadband filtered PD impulse ($f_1/f_2 = 100\,\text{kHz}/1\,\text{MHz}$)

(**c**) Narrow band filtered PD impulse ($\Delta f = 30\,\text{kHz}$, $f_m = 600\,\text{kHz}$)

**Figure 2.** Influence of the filter parameters when filtering a transient signal with a Butterworth filter.

## 3. Measurement Setup

The measurements were carried out with a one stage lightning impulse (LI) generator (Figure 3), in order to gain reproducible results [8]. The setup itself was based on a previous work [9], where PD in insulation oil under LI voltages up to 100 kV were investigated. For the purpose of this study, a form coil consisting of two parallel copper conductors with a mica based winding insulation was used.

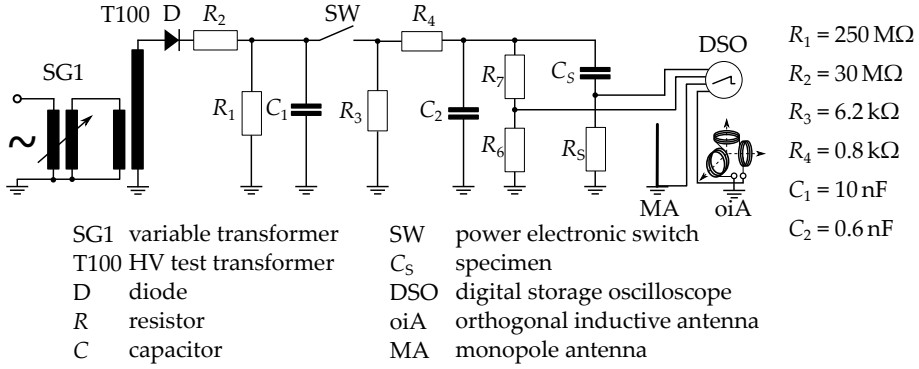

| | | |
|---|---|---|
| SG1 | variable transformer | |
| T100 | HV test transformer | |
| D | diode | |
| R | resistor | |
| C | capacitor | |

| | |
|---|---|
| SW | power electronic switch |
| $C_S$ | specimen |
| DSO | digital storage oscilloscope |
| oiA | orthogonal inductive antenna |
| MA | monopole antenna |

$R_1 = 250\,\text{M}\Omega$
$R_2 = 30\,\text{M}\Omega$
$R_3 = 6.2\,\text{k}\Omega$
$R_4 = 0.8\,\text{k}\Omega$
$C_1 = 10\,\text{nF}$
$C_2 = 0.6\,\text{nF}$

**Figure 3.** Circuit diagram for the measurement setup.

In order to achieve a reproducible rise time for one digit test voltages, the conventional sphere spark gap was replaced by a self-developed power electronic switch and was added. This was necessary,

because the commercially available, tested switches were found to produce disturbances by the internal DC/DC-converters, and a PD-free, thyristor based switch was developed and successfully applied. A high ohmic serial charging resistor $R_2$ was added to limit the final current. The thyristor cascade switch would then shut off independently. The utilized impulse voltage generator was PD-free in itself and produced a repeatable, stable voltage impulse. Thus, it allowed correct PD detection and measurements, due to a fingerprint-like impulse current signal caused by the voltage impulse. The PD signals were detected by a broadband 50 Ω-shunt, monopole antenna with a cut-off frequency $f_c$ of:

$$f_c = \frac{c}{4 \cdot l} \approx 75\,\text{MHz} \tag{1}$$

and an orthogonally wound inductive sensor.

The four signals (voltage measurement divider, shunt, antenna, and inductive sensor) were recorded synchronously with a four channel oscilloscope (Tektronix DPO 7104, bandwidth = 1 GHz, memory depth = 250 MSamples, vertical resolution = 8 bit).

The shunt value was chosen based on the measuring sensitivity and the impedance match of the coaxial RG85 cable. Furthermore, it needed to have a linear transmission behavior for frequencies of up to 100 MHz.

In order to suppress the impulse voltage caused by the current component while yielding a significant signal amplitude, a combination of a fifth order high pass filter and linear amplifier with $G = 10$ was used.

Primarily, the rise times of the different components needed to be recorded precisely to distinguish between them in the frequency domain.

The variable and the test transformer charged the capacitor $C_1$ while the power electronic switch SW was turned off. To generate a lightning impulse, the power electronic switch was turned on for $t_{\text{on}} = 500\,\mu\text{s}$, thus allowing $C_1$ to discharge fully over $R_3$, $R_4$, $C_2$, and the specimen $C_S$. The value of $R_1 =$ needed to be sufficiently high so as to prevent a discharge over it. The diode D protected the transformers against feedback effects. By using a power electronics switch instead of a spark gap, a clean and stable LI was generated without any ripples in the form (Figure 4).

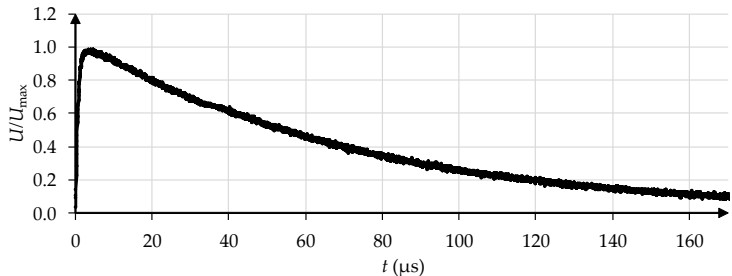

**Figure 4.** LI generated by the test setup ($T_1 = 1.96\,\mu\text{s}$, $T_2 = 54.23\,\mu\text{s}$).

## 4. Results

### 4.1. System Response

When an LI was applied to the specimen, all sensors measured a signal. Depending on the characteristics of the sensors, the measured signal differed. The reaction of the system is called system response and varied for all sensors (Figure 5).

The signal measured on the shunt in combination with a filter showed a very short impulse, which lasted approximately 1 µs. The displacement current was almost fully suppressed by the filter. The coil antenna in combination with the filter showed a slight overshoot of the initial impulse, which led to a duration of the impulse of several microseconds. For the monopole antenna, the initial impulse led to an oscillation of small amplitude, which lasted approximately 17 µs. Furthermore, it detected an interfering signal, which originated from the power electronic components of the

switch at ≈79 µs. The filters used in combination with the shunt and the coil antenna suppressed the interfering signals successfully (Figure 5).

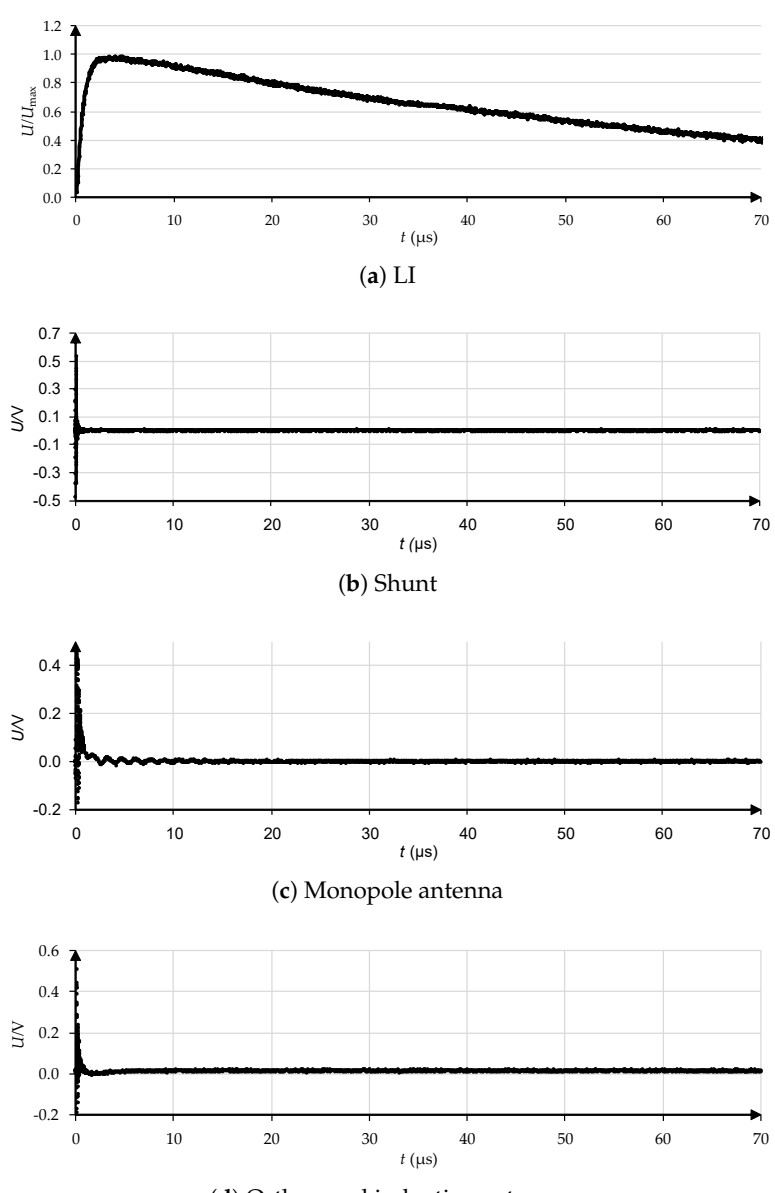

(**a**) LI

(**b**) Shunt

(**c**) Monopole antenna

(**d**) Orthogonal inductive antenna

**Figure 5.** System response of the different sensors to an LI.

## 4.2. PD Detection

The following oscillograms (Figure 6) showed an exemplary measurement result for PD inception at the test arrangement: the test voltage is plotted in temporal correspondence with the signals of the shunt, the monopole, and the coil antenna.

The shunt signal delivered a single peak just after the voltage peak. This signal was also detected by the monopole and coil antenna, while the amplitude of the impulse signals corresponded to the distance of both sensors. The monopole antenna yielded a lower signal amplitude because of the far off placement, but it showed a high sensitivity for disturbances of the surrounding.

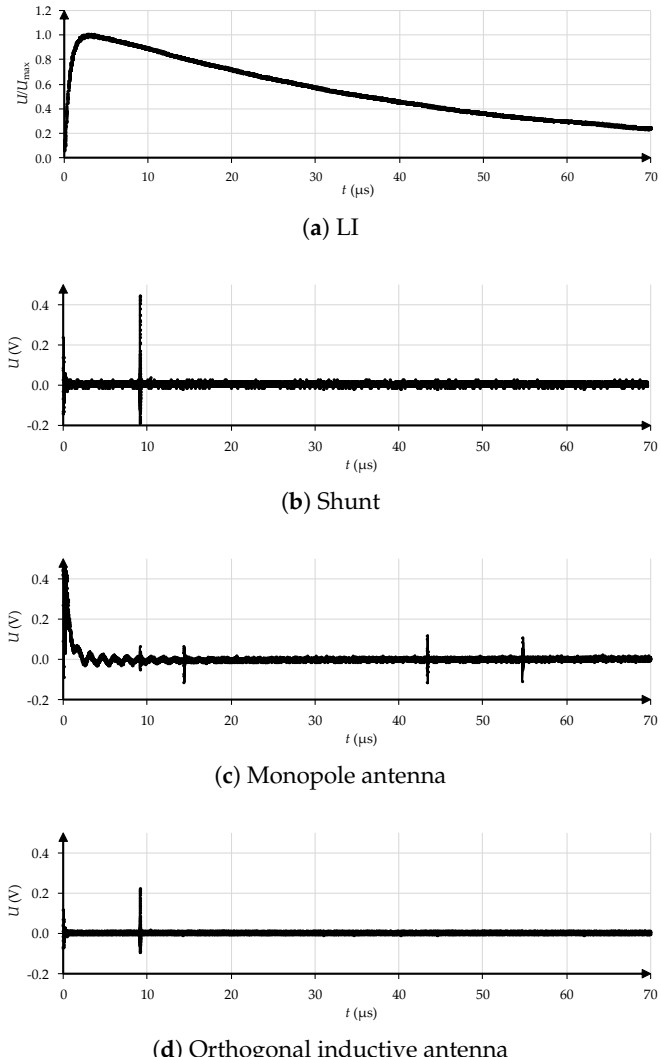

**Figure 6.** PD detection signals under impulse voltage stress.

The measurement system enabled the reliable detection of PD under impulse voltage stress (Figure 6). The first impulse of each sensor could be disregarded, as it is known to be the respective system response. All three sensors detected a PD impulse at $t \approx 9\,\mu s$. The monopole antenna furthermore detected several other impulses, which were not depicted by the other two sensors. These were interfering signals, which were suppressed by the filters used in combination with the coil antenna and the shunt, and thus no PD (Figure 6).

A closeup look at the first two signals aside from the system response of the monopole antenna confirmed a qualitative distinction between the PD and the disturbance signal (Figure 7a,b). The first peak of the disturbance was positive and smaller than the second one. After the maximum, the amplitude of the peaks decreased continuously. The oscillation lasted approximately 80 ns. The PD signal differed from the disturbance pattern as follows: The first peak was negative, and the following oscillation did not decrease continuously. The signal lasted for approximately 210 ns. A possible explanation for the disturbances was that the monopole antenna captured the incident wave and a multiple of its reflections. Metallic structures in the vicinity of the test setup could lead to reflections.

Furthermore, the system response of the shunt, as well as the PD detected by the shunt are displayed in a close up manner. A clear distinction is possible. While the system response oscillated for more than 400 ns, the PD signal lasted for less than 100 s. Furthermore, the system response displayed several peaks of similar amplitude, whereas the PD consisted of one peak with a significantly higher amplitude.

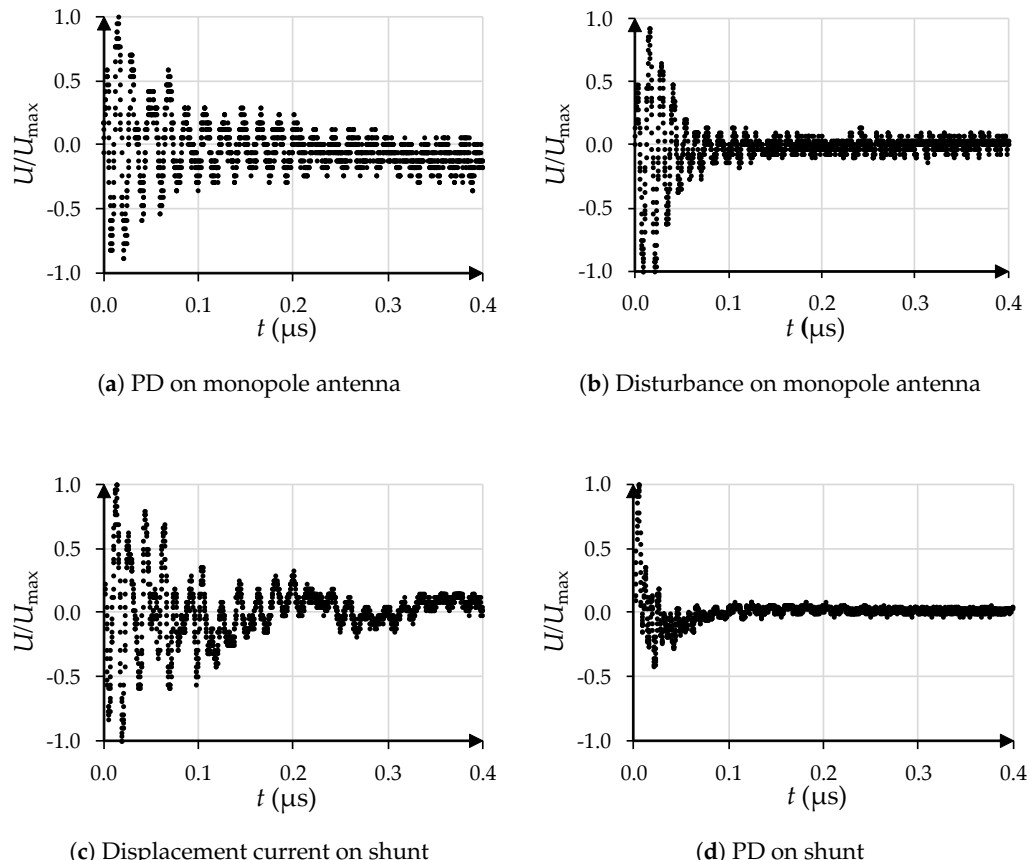

(**a**) PD on monopole antenna　　　　　　　　　(**b**) Disturbance on monopole antenna

(**c**) Displacement current on shunt　　　　　　　　　(**d**) PD on shunt

**Figure 7.** Detail of the normalized signal components detected by the monopole antenna and the shunt.

### 4.3. Application of Different Filters

The original acquired signals had a high ringing content. Therefore, as the next step, they were post-processed by applying digital Butterworth filters via MATLAB. As a kind of benchmark, the common filters known from IEC 60270, UHF, and VHF were chosen in order to compare the outcome, as it would provide a benefit for future separation techniques (Figure 8).

In the first place, it was very obvious that the filters altered the original signals many-fold: First, there was a significant reduction in amplitude from volts to millivolts caused by all filters. Both the IEC 60270 band filter (BF) and the 5 MHz low pass filter (LPF) clarified the the signals from the high frequency ringing. Because of the band width limitation, the filter had a significantly longer duration than the original signal, which might lead to uncertainties for highly repetitive events. Both the VHF and the UHF high pass filter (HPF) suppressed the signal content, which could be used for separation and leave only high frequency ringing.

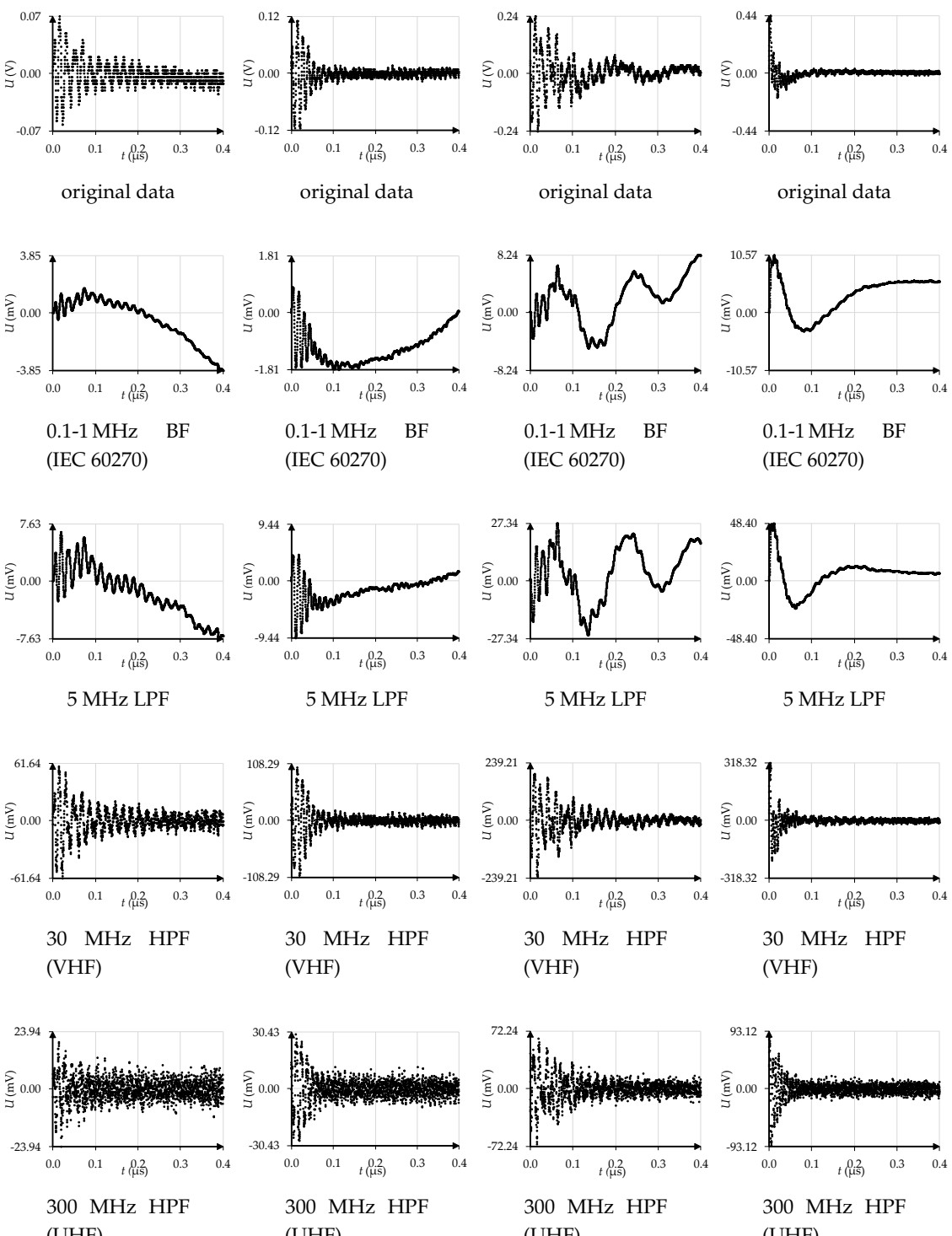

**Figure 8.** From left to right: PD on monopole antenna, disturbance on monopole antenna, displacement current shunt, PD shunt.

## 5. Discussion

The working thesis to detect PD events in an unshielded laboratory environment by identifying disturbances with the help of additional antennas was verified. For precise PD measurements under impulse voltage stress, it was seen as essential that in fact the applied test voltage needed to be very reproducible in terms of amplitude and time parameters for its effects on the current signals gained

from the test specimen. This can only be achieved by power electronic switches. Care needs to be taken that the test circuit itself, especially the switch, is free of PD or other sources of interference.

It could be shown that the PD detection for LI voltage stress was possible utilizing conventional measurement equipment.

The application of the three different sensors allowed a high confidence in the measured results. Especially, the self-constructed orthogonal inductive antenna showed in combination with the applied filter a high sensitivity to PD events and low sensitivity for distortion. The investigation showed that the UHF or VHF frequency range known from the measurements in power transformers [10,11] might be inadequate to detect PD in solid–air insulation systems. Neither frequency ranges offered the possibility to distinguish clearly between desired PD signals and disturbance. Although the frequency components occurring were clearly higher compared to IEC 60270, the frequency band appeared appropriate to yield a signal transformation necessary for a clear separation of different events. In order to perform a digital data post-processing, distinct requirements exist for the recording oscilloscope in terms of frequency range and storage depth.

The presented event oscillograms did not claim general validity. They were specific regarding the setup, sensors, and filters occurring. The quality of the signals may easily change once different filters or antennas are utilized; see [12,13]. However, due to the different path or coupling of the PD, respectively, into the sensors, the signal would always be different compared to an external distortion.

In order to be able to reliably detect PDs, even in a disturbance prone environment, further investigations are necessary. As shown in Figures 2 and 6, PD identification solely on the basis of their spectrum is not reliable. An identification of PDs based on their characteristic properties and course is an approach for better detectability in noisy signals.

## 6. Conclusions and Outlook

The paper presented a successful method to gain sound PD detection measurement results for LI voltage along with a disturbance separation. This was done first by comparing the amplitude and time based occurrence of the signals provided by the different sensors. Furthermore, differences in the oscillation patterns by respective sources were identified. It was therefore possible to diminish disturbances from signals caused by PD events. This would enable authors to accompany PD measurements for planned aging trials in unshielded laboratory environments with arbitrary voltage sources in the future. The aging of various insulation systems with bipolar damped oscillated impulse as investigated in [3] will be the subject of future research.

The measured signal of a single PD event was seen to contain even more information when considering the PD avalanche itself as a sender, the random dispersion area of the resulting electromagnetic wave as a non-uniform filter, and the sensors as receivers. The occurrence of outer corona discharges, surface creepage discharges, or the inner PD thoroughly eroding the insulation eventually must emit different electromagnetic pulses. Considering the insulation system of an electric induction motor, different PD will possess different antenna factors due to the different locations in relation to the metal components and the wave propagation. The authors will continue to develop the measurement technique further. The findings of the method will allow a sophisticated numerical evaluation of the measurement data. Wavelet methods such as in [14–16] will be taken further in order to distinguish between PD signals, noise, and disturbance automatically. Applications of alternative methods used for high DC voltage such as [17,18] are feasible, as well.

**Author Contributions:** Conceptualization: K.B.; methodology: K.B.; validation: K.B. and K.P.; formal analysis: L.E.; investigation and measurements: L.E. and K.B.; resources: K.B.; data curation: L.E.; writing, original draft preparation: L.E., K.B.; writing, review and editing: K.P.; visualization: L.E., K.B.; supervision: K.B.; project administration, K.B.; funding acquisition, K.B.

**Funding:** The authors gratefully thank VEM Sachsenwerk GmbH, Dresden, for funding this research.

**Acknowledgments:** Open Access Funding by the Publication Fund of the TU Dresden.



**Conflicts of Interest:** The authors declare no conflict of interest. The funders had no role in the design of the study; in the collection, analyses, or interpretation of data; in the writing of the manuscript; nor in the decision to publish the results.

## Abbreviations

The following abbreviations are used in this manuscript:

| | |
|---|---|
| BF | bandpass filter |
| DC | direct current |
| HPF | high pass filter |
| LI | lightning impulse according to IEC 60060-1 |
| LFP | low pass filter |
| MDPI | Multidisciplinary Digital Publishing Institute |
| PD | partial discharge |
| PWM | pulse width modulation |

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
