# Peer review of "Detection Technique of Partial Discharges at Impulse Voltage with Appropriate Filter Settings for Signal Separation"

_energies, doi:10.3390/en12234445_

Round 1

Reviewer 1 Report

The title of the work, which is very extensive, does not correspond to its content.
Neither abstrac nor the introduction contain any mention of the purpose of the work, which would be more general than a mere examination of the effectiveness of PD recognition by two electromagnetic field sensors. what constitutes a necessary novelty of the published paper was not included. There was no presentation of the works of other teams conducting research on the occurrence and methods of PD recording, which, to my knowledge, are very numerous. For example, many teams are involved in the PD phenomenon in power transformers, where many methods of measurement and analysis have been developed to identify, locate and even classify PDs.
It is not clear whether the object is solid or solid-air insulation?
It is also not true that the method of e.g. acoustic or optical emission does not allow for quantitative evaluation of PD (as it is written in line 61). Here again, we recommend literature from the PD area. The article contains the so-called "obvious obvious", and does not introduce significant novelties in the development of PD methodologies. Section 2.1 is redundant because there is no section 2.2.
The article should contain a broader description of the "state of the art", the purpose of the research and additional results, at least the analysis, on the basis of which one could confirm the hypothesis that it is possible to use these sensors in a given system. The time course itself is too simplistic and there are many better methods of analysis - again, I refer to the literature on the subject.

Image 1 is a quotation [7]. Are OX values normalized? Sources should be in English, not every reader knows German. Why is the amplitude on the graph for PD zero for UHF when UHF is widely used fo PD detection? This is not clear and should be described in more detail.
For a reader it is not/ may not be known what "LI" is, please explain it better. Line 101 speaks of a frequency spectrum, but there are no spectrograms in the work.
I recommend to perform additional analyses, which would illustrate in a collective way the sets of measurements and in a qualitative and quantitative way would confirm the hypothesis, which is also included in the title of the paper. Perhaps the wavelet methods suggested in section 6?

Author Response

Dear Reviewer,

I thank you for constructive critics of our contribution. I would like to respond to some of your remarks as follows:

Purpose of work ... Abstract and Introduction have been edited.

Novelty ... The paper presents the current state of the groups work. Possible novel thoughts are addressed in the outlook to point out our future work.

Work of other researcher ... of course there are a numerous groups doing research in the field of PD. We do not use the UHF method used for transformers, for it does not work with our scope due to the low signal amplitudes.

When using a lower and wider frequency band, for example Hammerström would probably have found out in 'Partial Discharge Characteristics within Motor Insulation exposed to Multi-Level PWM Waveform' (2018, IEEE ToDaEI) that the Behlke switch he used is producing already disturbance signals once connected with power supply.

Investigated object … form coil, both solid and air-solid insulation. Edited in document

Near-zero amplitude for PD in UHF … this why UHF method need high gain amps in order to get mV signals.

LI is the standard abbreviation by IEC 60060 for Lighting Impulse.

Kind regards,

 Karsten Backhaus

Reviewer 2 Report

authors are not aware of the most recent work on this field (as shown also by references). Measuring PFD under slow voltage impulses is not only carried out already, but even standardized (see  e.g. ref. [4], IEC 60034-27-5, Rotating electrical machines - Part 27-5: Off-line measurement of partial discharge inception voltage on winding insulation under repetitive impulse voltage, ). The  same will hold for oscillating waves, which is the next step for the authors. The real challenge is repetitive voltage impulses with very fast rise time (ns range), which is now faced by the most recent research.

In addition to the above concerns, there are sentences which are incomprehensible, see e.g. that between lines 25 and 28, other that are wrong, as “In contrast to square impulse, the bipolar damped impulses deliver quicker33 test result due to the higher number of zero crossings or trains of Upp-events” (note the opposite holds, because repetitive impulse frequency can go up to several kHz, as it is in reality in SiC converters).

Author Response

I thank you for the critical review of the article. I would like to answer your comments as follows:

“authors are not aware of the most recent work on this field (as shown also by references).”

I allow myself to disagree on your statement. For I follow many studies carried out in this field. In my point of view, there are a lot results of measurements out there, but without detailed information of fundamental setups and used equipment. This is what I want to do diffently.

“Measuring PFD under slow voltage impulses is not only carried out already, but even standardized (see  e.g. ref. [4], IEC 60034-27-5, Rotating electrical machines - Part 27-5: Off-line measurement of partial discharge inception voltage on winding insulation under repetitive impulse voltage, ).”

Of course, there is a standard with IEC 60034-27-5 for there is a high need by industry. Unfortunately, in my point of view there is no common opinion about, how to do the measurements right. The IEC 60270 is even older. Still there is no OEM provind a coupling device with a constant transfer amplitude spectrum as required by the standard. All devices I tested myself show Eigen oscillation effects in the relevant frequency domain, which contradict with the requirements for the ‘quasi integration’. Please name me an OEM who fulfils that.

For impulse voltage PD measurement it is a similar case. When comparing results of OEM equipment, which I am not allowed to publish, one will get different results for the same test object. That is why the standard requires noting the method used. And that is also why I chose the fundamental approach, stating explicitly all components of the setup.

“The same will hold for oscillating waves, which is the next step for the authors. The real challenge is repetitive voltage impulses with very fast rise time (ns range), which is now faced by the most recent research.”

I fully agree on your estimation regarding fast ns impulse switching. Again, I would like to get the µs-switching right in the first place and then reducing the orders of the rise times.

“In addition to the above concerns, there are sentences which are incomprehensible, see e.g. that between lines 25 and 28, other that are wrong, as “In contrast to square impulse, the bipolar damped impulses deliver quicker33 test result due to the higher number of zero crossings or trains of Upp-events” (note the opposite holds, because repetitive impulse frequency can go up to several kHz, as it is in reality in SiC converters)”

The damped oscillating impulse generator developed in Dresden and used by my former colleague Davoud Moghadam, delivers your stated requirement of zero crossings in the range of many kHz by a fundamental one-digit-kHz-basic-repetition rate.

Reviewer 3 Report

The author well documented this manuscript and should go through the manuscript once to fix the typos.

Author Response

Thank your for your intercession!

Round 2

Reviewer 1 Report

1. In my opinion, the title should be written more precise because it is too broad, whereas the content only concerns switches.
2. I would advise that the article should be extended to include the results of additional studies at least in the frequency domain. In its current form this is sufficient for a scientific conference, but too trivial for a high IF journal like Energies.
3. there is no information on how many measurements have been made, what is the repeatability of the method? evaluation of the possibility of using the method on the basis of a single chart and in the time domain is, in my opinion, too trivial today, especially since there are many easily accessible tools for signal analysis, e.g. MATLAB, which is used by the authors.
4. the list of references is quite poor (16 items), the first reference is PhD thesis from 1995 (24 years old) in German and additionally one more book in German, is there no literature in English? The 5 literature items are only mentioned as possible sources of knowledge in the summary.
5. The introduction should still refer in more detail to the method used, with an indication of the literature on the subject.
6. Section 6 ends in a bizarre way: "The authors will there ..."

Author Response

Respected Reviewer,

beginning with the new title and therefore changing the scope of the paper we reworked the complete manuskript. There is an additonal paragrapf on data post processing in order to make our approach more clearly.

All authors looking forward to your feedback.

Kind regards,

Ba El Pa

Reviewer 2 Report

While I respect and appreciate the author answers, I keep not considering the paper ready for publication.

Author Response

(The authors gave the same response as above.)

Reviewer 3 Report

The authors have given enough details in the manuscript.

Author Response

Dear Reviewer,

after the iterations with the other reviewers, there are significant changes / added contenct to the manuscipt.

Kind regards,

 Karsten Backhaus

Round 3

Reviewer 1 Report

The authors have done their best to improve the article, so after they have corrected a few punctuation mistakes, I recommend the article for publication in Energies.

Please delete the "dot" in line 7: "stress. by" Please change beginning of the sentence in line [29]: "[1] concluded (...)"  - "Authors in [1] concluded", or sth similar.  Same for line 34: "[3] determined (...)". Please delete the ")" which is doubled by reference to Fig.2b and Fig.2c. in line 96. and the same for line 155 "(Fig. 7 a) and b))"  Please add a dot at the end of sentence in line 113. line 219: ".. must emit different electromagnetic pluses.."  --> pulses.

Author Response

Dear reviewer,

I thank you very much for your constructive criticism, that lead to the good state of the paper as it is. I worked your final suggestions into the manuskript.

I am looking forward to the groups upcoming results with the proposed methods. I am planning to publish them on MDPI again. Perhaps you will then take again the effort of review.

Kind regards,

 Karsten Backhaus

Reviewer 2 Report

In spite of revisions, I do not still feel comfortable with this paper.

Author Response

Dear reviewer,

I thank you very much for your constructive criticism, that lead to the good state of the paper as it is.

Kind regards,

 Karsten Backhaus
